# Polymeric Excipients in the Technology of Floating Drug Delivery Systems

**DOI:** 10.3390/pharmaceutics14122779

**Published:** 2022-12-12

**Authors:** Evgenia V. Blynskaya, Sergey V. Tishkov, Vladimir P. Vinogradov, Konstantin V. Alekseev, Anna I. Marakhova, Alexandre A. Vetcher

**Affiliations:** 1V. V. Zakusov Research Institute of Pharmacology, 8 Baltiyskaya St., 125315 Moscow, Russia; 2Institute of Biochemical Technology and Nanotechnology, Peoples’ Friendship University of Russia (RUDN), 6 Miklukho-Maklaya St., 117198 Moscow, Russia; 3Complementary and Integrative Health Clinic of Dr. Shishonin, 5 Yasnogorskaya St., 117588 Moscow, Russia

**Keywords:** polymers, matrix formers, gastroretentive drug delivery systems, floating dosage forms, buoyancy

## Abstract

The combination of targeted transport and improvement of the release profile of the active pharmaceutical ingredient (API) is a current trend in the development of oral medicinal products (MP). A well-known way to implement this concept is to obtain floating gastroretentive delivery systems that provide a long stay of the dosage form (DF) on the surface of the stomach contents. The nomenclature of excipients (Es) of a polymeric nature used in the technology of obtaining floating drug delivery systems (FDDS) is discussed. Based on the data presented in research papers, the most widely used groups of polymers, their properties, and their purpose in various technological approaches to achieving buoyancy have been determined. In addition, ways to modify the release of APIs in these systems and the Es used for this are described. The current trends in the use of polymers in the technology of floating dosage forms (FDF) and generalized conclusions about the prospects of this direction are outlined.

## 1. Introduction

The oral administration route has a wide range of advantages, including ease of use, high patient compliance, flexibility in the formulation of dosage forms, ease of storage and transportation, no need for specially trained personnel, etc. [1,2,3]. According to reviews on this topic, more than 50% of drugs available on the market must be taken per os [4,5]. However, most oral drug delivery systems are subject to physiological factors that can adversely affect the bioavailability of the API and the clinical efficacy of the drug, such as the pH value of the medium, the heterogeneity of absorption throughout the gastrointestinal tract (GIT), the difference in surface area and enzymatic activity of various sections of the gastrointestinal tract, transit time through the sections in which absorption occurs, and the rate of metabolism and excretion [1,2,3,6]. Due to the high significance of the described effects on the action of many drugs, two main strategic directions for improving oral dosage forms have been proposed: targeted delivery and modified release of API.

The representatives of the described approach include the gastroretentive delivery systems that provide the release of drugs in the upper gastrointestinal tract for a long time on a controlled basis [1,2,3,6,7]. The combination of the described properties makes it possible to improve the bioavailability of drugs that are absorbed mainly in an acidic environment or have a narrow “absorption window”, to ensure a decrease in fluctuations in the concentration of API in the blood plasma, and also makes it possible to increase the effectiveness of drugs that act locally in the stomach [1,2,3,6,7]. The clinical advantages of gastroretentive systems are: reduction of fluctuations in the therapeutic effect for drugs with a concentration-dependent effect and prevention of the development of concentration-dependent undesirable effects, prolongation of the maintenance period of therapeutic concentration for drugs with time-dependent pharmacodynamics, prevention of undesirable activation of counterregulatory mechanisms, rebound effect and tolerance, reduction of undesirable MP effects, by reducing the entry of API into other parts of the gastrointestinal tract [1,2,3,6,7]. To date, several main types of gastroretentive systems are known, such as magnetic, mucoadhesive, high-density, raft-forming, floating, and increasing in size, as well as superporous hydrogels [2]. Among the described variations, the most reflected in research papers were FDF, which favorably differ in independence from intraspecies differences (the size of the pyloric sphincter) and the state of the mucous membranes of the gastrointestinal tract, and also have a comparative ease of preparation, which implies the use of standard technological processes [1,2,3,5,7,8,9,10,11]. When obtaining all the mentioned types of gastroretentive delivery systems, and especially floating ones, it is typical to use various Es of a polymeric nature, which ensure the implementation of one or another approach to retention in the stomach. As a result, it is necessary to carefully select the appropriate components of the pharmaceutical composition to control the technological, pharmaceutical, and pharmacokinetic properties of DF.

The purpose of this article is to identify the most common types of polymers used to obtain FDF, to analyze their properties and purpose in relation to each technological approach and release modification method, as well as to establish promising directions for the search and selection of new materials for further developments in this area.

### Methodology for Preparing the Review

In order to prepare a review of the use of polymeric Es for the production of FDDS, applied articles describing the composition and technology for obtaining FDF published from 2013 to 2022 were selected, for which a search was made in the databases Scopus, PubMed, and Web of Science for the following keywords: gastro-retentive, gastro retentive, gastroretentive, gastroretentive formulations, gastroretentive drug delivery systems, gastroretentive dosage forms, gastric retention, gastroretention, gastroretentive, gastroretentive systems, floating, floating dds, floating dosage forms, floating dosage form, floating drug delivery system, floating drug delivery, floating tablets, floating beads, floating microspheres, floating capsules, floating systems, buoyant dosage form, gastric-floating drug delivery systems, and gastric floating. 

Articles were included in the review based on the following criteria: obtaining an FDDS (or a delivery system with flotation as one of the mechanisms of gastroretention), assessment of buoyancy properties (floating lag time and total flotation time) using conventional methods (or if there is justification for the use of applied evaluation techniques), a description of the purpose of the polymeric Es in the pharmaceutical composition (or an indication of the dependence of the properties of the dosage form on their presence or amount). The exclusion of publications was carried out on the basis of non-compliance with their inclusion criteria, as well as when describing the raft-forming system as an FDDS. Initially, 139 publications were selected for review.

## 2. Floating Drug Delivery Systems

Floating drug delivery systems, DF, the value of the bulk density of which is initially lower than the density of gastric juice (1.004 g/cm^3^) or decreases to the required value after ingestion for a certain period of time [1,2,3,6,7]. This property allows the drug to remain on the surface of the contents of the stomach, thereby avoiding evacuation to the underlying sections of the gastrointestinal tract, without affecting the rate of emptying and without harming the mucous membranes [1,2,3,6,7]. Flotation provides the system with targeted delivery with the ability to release the API in the required way for a time corresponding to the clinical need [1,2,3,6,7].

## 3. The Place of Polymers in the Technology of Floating Drug Delivery Systems

As mentioned above, the main advantages of floating gastroretentive systems are targeted delivery and the modified release of the API. Each of these properties is most often realized through the use of certain Es, in most cases of a polymeric nature (Figure 1).

Targeted drug delivery in this case manifests itself in the form of a buoyancy property achieved in various ways, depending on the technological approaches that traditionally underlie the FDF classification. At the moment, two main directions are distinguished among these: obtaining systems with an initially low density, by incorporating air or explosives with a low bulk density, and obtaining systems density, which decreases after ingestion due to swelling or gas formation (Figure 2) [2].

The initially low density of DF, as mentioned above, can be achieved in one of two ways: the first is the creation of air pores or cavities, and the second is the inclusion of special Es. Air incorporation is achieved by freeze drying, sublimation of pre-introduced volatile components (for example, camphor or menthol), extrusion, sonication of molten intermediates, creation of hollow microspheres, or 3D printing of delivery systems with pre-designed cavities [2,3]. This approach provides zero ascent delay time, reducing the risk of the premature evacuation of MP from the stomach [2]. The described methods, as a rule, are complicated or require the inclusion of additional stages in the technology for obtaining such DF as tablets, granules, pellets, etc. Polymers in this approach to achieving flotation are used to create a matrix or shell to hold gasses within the system. The second option, based on the inclusion of low-density Es, also makes it possible to ensure the instantaneous ascent of the DF, however, it allows the use of standard technological processes. For these purposes, the use of oils or lipophilic substances, water-insoluble fillers with a low bulk density, such as ethylcellulose (EC), is common, and the use of foamed polypropylene is also mentioned in the sources [2,3,12,13,14]. The range of application of polymers in this approach is the production of a shell or matrix holding Es with a low density; in some cases, these substances can themselves act as an agent that provides buoyancy.

The decrease in the density of the floating delivery system when it enters the stomach is achieved either by increasing the DF in volume disproportionately to the increase in its mass, or by producing gas. As a rule, the delay time of the ascent of these drugs is several minutes, negatively affecting the risk of entering the lower gastrointestinal tract; however, methods for obtaining such FDF are usually simple and do not require deviations from standard technologies. The swelling of the system occurs due to the addition of hydrophilic matrix-forming polymers that are able to retain their shape and erode for a long time [1,2,3,12,13]. The gas formation approach is based on the application of the reaction between carbonates (sodium bicarbonate, sodium bicarbonate, calcium carbonate) and acids (citric, tartaric, and others), which make up a special mixture, and the acidic contents of the stomach [1,2,3,12,13]. The emitted CO_2_ is retained by a matrix or shell, for which polymeric Es are used. In addition to the one described above, there is theoretically a concept of reducing the density of the system due to the phase transition of a liquid evaporating at body temperature (ether or cyclopentane) placed in a reservoir inside the DF [1,2,3,15,16]. In this embodiment, polymeric Es are used to create a chamber and a biodegradable plug that make up the described reservoir.

All approaches, as a rule, are universal for most dosage forms, among which there are single-component systems: tablets, capsules, sponges (foams); and multi-component systems: microcapsules, granules, pellets, and others, however, the suitability of certain methods for them varies (Figure 2) [1,2,3,7].

The modified release is a characteristic feature of floating delivery systems and is usually implemented in one of two ways; controlled release or pulsatile release [1,2,3,6,7]. In the first case, the Es of a polymeric nature act most often as matrix-forming substances, less often in the form of coatings. In the second, DF, in most cases, is a structure consisting of several layers, which can be thin films or, for example, bulk-pressed coatings. Based on the data presented above, most often the polymers used to ensure the buoyancy of the system, and, at the same time, have properties that provide a modified release of the API.

Like any other approach to gastroretention, floating systems have certain limitations related to the amount of gastric juice, the acidity of the environment, food intake, and ascent delay time [1,2,3]. Therefore, a combinatorial approach has become widespread, which implies a combination of flotation, most often with mucoadhesion or an increase in the size of the system, which makes it possible to reduce the risk of premature DF evacuation into the lower gastrointestinal tract [1,2,3]. However, in this case, it is necessary to use either special Es to provide additional properties, or Es that combine them with the characteristics necessary for implementing approaches to obtaining floating systems. Many Es used in FDF technology, such as hydroxypropyl methylcellulose (HPMC), chitosan, carbomers, polyethylene oxide (PEO), calcium pectinate, xanthan, have mucoadhesive properties. A significant increase in size during swelling can be achieved with guar gum, HPMC, and sodium carboxymethyl cellulose (NCMC), which are also used to achieve buoyancy.

Based on the purpose of polymeric excipients in the field of obtaining floating delivery systems, the choice of Es is determined by the technology for obtaining DF and such basic characteristics as solubility in water and organic media, hydrophilicity, lipophilicity, the ability to swell in water, gel strength, and density.

## 4. Polymers of Natural Origin Used in the Technology of Floating Delivery Systems

The use of individual Es of a polymeric nature in the technology of floating systems, as a rule, is not limited to a certain approach to achieving buoyancy. Therefore it is convenient to subdivide them into groups according to their origin: natural origin, semi-synthetic, and synthetic.

Polymers obtained from natural sources in the FDF technology are most often represented by polysaccharides, such as sodium alginate (NA), chitosan, pectin, gums, etc. The use of these substances is a progressive “green” approach in pharmacy, which consists in the popularization of Es of natural origin, but in most cases, it is common in research work to find new excipients or substitutes for existing ones. As a rule, the described polymers have advantages in terms of swelling index and hydration rate but are subject to changes in technological characteristics during storage, for example, viscosity, loss of residual moisture, degree of contamination by microorganisms, ability to provide controlled release, and some other properties [17,18].

### 4.1. Alginates

Sodium and calcium salts of alginic acid have found wide application in the technology of floating delivery systems. NA is a heteropolymer formed by polyuronic acid residues of G-fragments (α-l-guluronic acid residue) and M-fragments (β-d-mannuronic acid residue) [13,19,20]. This polysaccharide slowly dissolves in water with the formation of a viscous colloidal solution (viscosity of a 1% solution is 20–400 cP at 20 °C), is practically insoluble in solutions with a pH below 3, and has bioadhesive properties [1,2,3,19,21]. In FDF technology, it can be used as a matrix-forming substance; in addition, NA reacts with salts of Ca^2+^ and other metals, forming water-insoluble gels with a network structure, capable of adsorption and delayed release of API [13]. The described reaction is often used to obtain microcapsules by ionotropic gelation [2].

For example, Praveen et al., used NA to obtain granules by ionotropic gelation, which have a low density due to the porosity obtained by freeze drying, and the addition of myristyl alcohol as a low-density substance. DF floated up instantly, the duration of flotation was more than 20 h, and a prolonged release was achieved [22]. Falcone et al., used the ability of NA to react with calcium salts to prepare two types of inks for 3D printing, based on a matrix former and a crosslinking agent. The obtained DF, consisting of hollow filaments, demonstrated instantaneous ascent, and flotation for 14 h, releasing 80% of the drug during this time in a slow manner [23]. Using olive oil to reduce the density of the delivery system, Adebisi et al., obtained floating granules of clarithromycin based on NA by ionotropic gelation, coated with a mucoadhesive layer of chitosan. DF demonstrated flotation for 24 h with a controlled release of API for 8 h [24].

### 4.2. Chitosan

Chitosan is a linear aminopolysaccharide (copolymer of glucosamine and N-acetylglucosamine) [13,19]. This polymer is slightly soluble in water, practically insoluble in organic solvents, alkaline solutions (at pH about 6.5), soluble in acids with protonation of amino groups, the specific density of the material is 1.35–1.40 g/cm^3^ [19]. In the technology of floating delivery systems, it is used as a matrix-forming agent, a film-forming substance, to give DF mucoadhesive properties; due to its ionic nature, it is widely used to obtain microcapsules by emulsion cross-linking and ionotropic gelation [1,2,3,15].

Svirskis et al., developed and manufactured hollow floating and mucoadhesive granules with acyclovir based on chitosan, using the ionotropic gelation method. The resulting DF had a zero ascent delay time and, at the same time, the buoyancy of most of the granules remained for more than 3 h, providing a delayed release [25]. Praveen et al., obtained floating granules from chitosan by ionotropic cross-linking, bearing carvedilol as API. DF demonstrated flotation for more than 10 h and delayed release of drugs for 8 h [26]. A non-standard method for obtaining floating dosage forms was demonstrated by Budianto et al., by synthesizing a substance that is chitosan with a grafted polyvinylpyrrolidone (PVP) molecule. A hydrogel based on this substance containing API: amoxicillin, after the addition of pore formers, demonstrated a low ascent delay time, a flotation time of more than 180 min, and a delayed release of MP [27].

### 4.3. Carrageenan

Carrageenans are linear sulfate polysaccharides consisting of sulfated galactose and 3,6-anhydrogalactose, sugars connected by alternating α-1,3 and β-1,4 glycosidic bonds [19]. The polymers are soluble in water when heated (20 °C or 80 °C depending on the family) with a solution viscosity of 5 cP (at 75 °C) [19]. These polysaccharides are used in the FDF technology as a gelling agent and retardant, and can also be used to manufacture microcapsules by crosslinking [19]. 

Using carrageenan alongside NA, Zhang et al., obtained floating granules containing B. javanica oil by ionotropic gelation. Manufactured DF provided delayed release of API for 6 h. Flotation was provided due to the properties of MP and the high porosity of the granules themselves [28]. Also combining iota-carrageenan and NA in the composition, Abbas and Alhamdany produced floating microcapsules of enalapril maleate using the ionotropic gelation method. Buoyancy was achieved due to the porosity of the DF matrix and the addition of a blowing agent; the time of flotation and controlled release of API was 24 h [29].

### 4.4. Pectins

Pectins are high molecular weight polysaccharides consisting mainly of D-galacturonic acid residues connected by α-1,4 glycosidic bonds in the chain; acid and hydroxyl groups can be esterified [13,19]. These polymers can be used in the technology of floating delivery systems as matrix formers, in addition, due to the ability to interact with polyvalent metal ions, they are used for the manufacture of microcapsules by the ionotropic gelation method; after reaction with calcium ions, the cross-linked gel acquires mucoadhesive properties [2,13,15,19,30].

Alhakamy et al., developed and manufactured granules containing ellagic acid based on the reaction of pectin with calcium ions, using the method of ionotropic gelation. The resulting DF had a flotation time of 24 h and sustained release for 24 h [30]. Abouelatta et al., also used the ionotropic gelation method to obtain pectin-based floating cinnarizine granules, achieving DF flotation time of 8 h, an immediate ascent, and a zero-order release kinetics within 12 h [31].

### 4.5. Xanthan Gum

Xanthan gum is a polysaccharide consisting of a main chain of D-glucose residues linked by β-1,4-glycosidic bonds with side chains, which are two linked units of D-mannose and a D-glucuronic acid residue [13,19]. This polymer is soluble in water, insoluble in organic solvents, a 1% aqueous solution has a pH of 6.0 to 8.0 and a viscosity of 1200–1600 cP at 25 °C [19]. In the FDF technology, xanthan is used as a matrix-forming agent to achieve a prolonged release of API, in addition, the substance has bioadhesive properties and the ability to form foamy structures [13,19].

Xanthan gum has been used as a matrix former in the production of gas-forming propranolol floating tablets by Lavanya et al., The resulting gastroretentive system had an ascent delay time of about 150 s and provided a controlled release of API for 12 h, while maintaining buoyancy [32]. A co-processed excipient of xanthan gum and acacia gum was developed and applied by Budaya and Surini to produce floating famotidine gas-generating tablets. The studied mixture had a high swelling index and made it possible to achieve a matrix that formed a strong gel. The resulting tablets retained the ability to float for 24 h and provided a controlled release of API [33].

### 4.6. Guar Gum

Guar gum is a polysaccharide containing a linear chain of D-mannopyranosyl units linked by β-1,4 glycosidic bonds to D-galactopyranosyl units attached by α-1,6 bonds [13,19]. This galactomannan is soluble in water; when it enters cold or hot water, it immediately swells, forming a highly viscous, thixotropic sol; a 1% solution has a pH of the medium of 5.0–7.0 and a dynamic viscosity of 4860 cP; the dry substance has a density of 1.492 g/cm^3^ [19]. In the technology for the production of FDF, guar gum is used as a matrix-forming agent and retardant, as well as to provide mucoadhesion, moreover, due to significant swelling in water, the polymer can both reduce the density of the DF and provide an increase in its size to prevent removal from the stomach [13,19,21].

Guar gum mixed with xanthan gum was used by Dey et al., to create matrix gas-forming tablets with atenolol. The resulting DF retained the ability to float for 12 h in vitro, and was retained in the stomach of the test animals (rabbits) in vivo for more than 6 h, providing a delayed release of API [34].

### 4.7. Other Natural Polymers

The search for new Es of natural origin or new sources of obtaining already known materials is of scientific interest. Many publications of this nature are devoted to the production of polymers for further use in the technology of gastroretentive drug delivery systems.

For example, Hendrika et al., isolated pectin from a banana peel (Musa balbisiana ABB), which was then used to react with calcium chloride to obtain amoxicillin granules by ionotropic gelation. Buoyancy was ensured by the addition of a gas-forming agent (sodium bicarbonate) to the composition, the DF ascent delay time was 20 s, the flotation time was more than 12 h, and API was released in accordance with the Higuchi kinetics model for 5 h [35]. Liu et al., used hyaluronic acid as a gelling agent in the preparation of pills containing dihydromyricetin by extrusion with further freeze-drying. Buoyancy was provided by the addition of cetyl alcohol as a low-density agent and a blowing agent (sodium bicarbonate), the optimal DF provided flotation and delayed release of the API for 24 h [36]. Khoder et al., received floating granules of ciprofloxacin, using as a matrix a gel-conjugate of bovine serum albumin with NA, further treated with calcium chloride. This delivery system had a foamy structure, due to which the flotation time was up to 48 h, DF provided a delayed release of API for 3 h [37]. Mahor et al., isolated a polysaccharide based on D-xylose and D-glucuronic acid from the seeds of Mimosa pudica, the resulting substance was used as a mucoadhesive and retardant in the preparation of gas-forming famotidine microcapsules from NA by ionotropic gelation, in addition, chitosan was used to slow down the release of famotidine and enhance adhesion to mucous membranes. DF had a controlled release of API and the ability to float for 12 h [38]. Taranally et al., used a combination of pectin, xanthan gum, and locust bean gum to produce floating metoprolol tartrate granules by ionotropic gelation. Flotation was achieved due to the high porosity of the calcium pectinate matrix and the addition of a gas-forming agent; the flotation time was more than 12 h; DF provided a pulsatile release [39]. A non-standard solution was proposed by Xu et al., by producing a hybrid gel of starch and microcrystalline cellulose, which, due to the three-dimensional structure, after freeze-drying, had high porosity and low density. Tablets loaded with ranitidine, based on the obtained excipient, retained buoyancy for 24 h [40]. Kukati et al., made floating tablets based on a matrix of locust bean gum, buoyancy was achieved due to the formation of pores during the sublimation of camphor. DF did not have an ascent delay and retained flotation and zero-order release kinetics for 12 h [41]. Xanthan gum, together with a hydrocolloid isolated from Orchis morio, was used as a matrix former for metformin-containing floating flatulent tablets by Razavi et al., The resulting dosage form in vivo demonstrated a delayed release of API in the stomach for 12 h [42]. Xanthan gum together with a powder obtained from the seed kernels of Tamarindus indica, Razavi et al., used as matrix-forming substances to produce metformin gas-forming tablets. DF floated for more than 24 h and provided a sustained release of API for 12 h [43]. Galactomannan isolated from Caesalpinia pulcherrima, Thombre and Gide was used together with NA as a matrix former to obtain amoxicillin granules by ionotropic gelation followed by coating with chitosan to provide mucoadhesive properties. Due to the high porosity and the addition of calcium carbonate as a gas-forming agent, DF was able to maintain buoyancy for 8 h and provide a controlled release of API [44]. An unusual way to use cellulose was demonstrated by Svagan et al., producing furosemide-loaded foam from this polymer. This DF remained buoyant for 24 h and provided a sustained release of API [45]. Izgelov et al., ovalbumin was used as a matrix-forming agent in the preparation of gas-forming tablets containing a self-emulsifying system of cannabinoids. The resulting DF floated up within 0.8 min, possessed gastroretentive properties in vivo, provided a slow release of API and a significant increase in its bioavailability [46].

## 5. Semi-Synthetic Polymers Used in the Technology of Floating Delivery Systems

A group of semi-synthetic polymers used in the technology of floating dosage forms, mainly represented by various cellulose derivatives, such as Ethyl cellulose ether (EC), Hydroxypropyl methylcellulose (HPMC), Hydroxypropyl cellulose (HPC), and Sodium carboxymethyl cellulose (NaCMC). These Es are available, well studied, have a wide range of grades specially designed for various needs, and therefore are widely used to obtain floating delivery systems.

### 5.1. EC

EC is a long-chain polymer, the chain of which consists of β-anhydroglucose units with acetal bonds [13,19]. The polymer is practically insoluble in water, glycerol, propylene glycol, absorbs small amounts of water, dissolves in various organic solvents, and has a bulk density of 0.4 g/cm^3^ [19]. EC can be used in the technology of obtaining FDF to obtain water-insoluble coatings capable of providing a modified release of MP and retaining gas inside the system [19]. In addition to the described variant, this polymer is used to obtain microspheres, by solvent removal, and also as a density-reducing agent [13,19].

Kohli et al., used EC to prepare hollow repaglinide microspheres by solvent removal. The resulting DF provided first-order API release kinetics, zero ascent delay time, and retained buoyancy for 12 h [47]. Liu et al., used EC (10 cP) in conjunction with HPMC K4M to obtain a more resistant matrix in the manufacture of gossypol gas-forming tablets. DFs were able to float for 12 h, had an ascent delay time of less than 1 min, and had a delayed release of API [48]. The low density, hydrophobic properties affecting the release and penetration of water into DF, EC, and the ability of HPMC to form stable gels were applied by Kim et al., upon receipt of sodium ecabeta tablets by molding and freeze-drying. The resulting porous DF had zero ascent delay time, retained gastroretentive properties for 12 h in vivo, and provided delayed release of API [49]. Nashar et al., developed a composition and technology for obtaining floating microspheres of clarithromycin, in which EC provided the formation of a skeleton upon removal of the solvent, and HPMC E5 had a smaller particle size, a higher loading and release rate of the API. The resulting DF immediately floated up, was able to stay on the surface of the dissolution medium for 8 h, and provided a slow release of clarithromycin [50].

### 5.2. HPMC

HPMC is a partially O-methylated and O-(2-hydroxypropylated) cellulose [13,19]. The presented polymer dissolves in water, forming a colloidal solution; the viscosity of a 2% aqueous solution, depending on the brand, ranges from 3 to 100,000 cP [13,19]. Bulk density and density after compaction of hypromellose are 0.341 and 0.557 g/cm^3^, respectively, which is lower than the value of the bulk density of gastric juice, but the true density is high and amounts to 1.326 g/cm^3^ [12,19]. In the FDF application technology, it is used as a matrix-forming agent, as well as an agent providing mucoadhesion [2,3,19]. A matrix based on high-viscosity hypromellose grades is able to retain CO_2_ released during the reaction of a gas-forming mixture, and also swells significantly upon contact with a liquid, which is used in the production of non-gas-forming FDFs and gastroretentive delivery systems that increase in size [1,2,3,21].

Losartan delayed-release matrix floating tablets were developed by Shinde et al., The DF consisted of three layers: an immediate release core, an intermediate pressed coating based on HPMC E50, a floating pressed layer, which is a mixture of HPMC K4M and sodium bicarbonate, in which the polymer retains the released gas. The tablets had a rise time delay of 5 min, a flotation time of more than 12 h, and an API release delay of 6 h [51]. Theophylline tablets based on HPMC grades K100LV, K4M, and K100M matrices were obtained by Sungthongjeen et al., The formation of the hypromellose gel ensured the retention of the gas formed due to the reaction of sodium bicarbonate and citric acid, and the delayed release of the API. The resulting tablets with each type of HPMC, depending on the composition, provided a satisfactory rise time delay of less than 15 min, a flotation time of more than 480 min, and a delayed release of theophylline [52]. Rao et al., developed floating tablets containing cefuroxime axetil, the gas-retaining matrix of which was formed by the HPMC brand K4M. DF of the optimal composition demonstrated the Korsmeier-Peppas release kinetics model for 12 h, retained the ability to float for 12 h in vitro and 6 h in vivo, and had a rise delay time of 2 min [53]. Jadi et al., received propranolol floating tablets containing Compritol 888 ATO (glyceryl behenate) as a low-density excipient and HPMC K4M as a matrix former. The resulting DF had a zero rise delay time and a flotation time of 12 h, and provided a delayed release of API for 10 h [54]. Malladi and Jukanti have developed clarithromycin matrix tablets based on HPMC K4M, which float due to the presence of pores formed during the sublimation of camphor introduced into the composition. The optimized DF had a zero rise time delay of 10 s, a flotation time of more than 12 h, and a delayed release of API [55].

### 5.3. NaCMC

NaCMC is the sodium salt of polycarboxymethyl cellulose ether [19]. Carmellose sodium is a water-soluble polymer capable of absorbing significant amounts of water and acting as a gelling agent with a high swelling index, the viscosity of 1% aqueous solutions varies from 5 to 20,000 cP depending on the brand, the bulk density, and after compaction is 0.52 and 0.78 g/cm^3^ [19,21]. In the technology of gastroretentive systems, this explosive is used in the composition of matrices and ensures the rapid formation of a gel layer on the surface of the system and a significant increase in DF in size.

Venkateswarlu and Chandrasekhar used NaCMC together with HPMC as matrix formers and retardants in the production of floating tablets. The DF flotation was ensured by CO_2_ released upon contact with the dissolution medium; in addition, cetyl alcohol was present in the composition to slow down the release, which also reduces the density of the system. Tablets of the optimal composition demonstrated a rise time delay of 8 s, a flotation time of more than 12 h, and a sustained release of 8 h [56]. Rapolu et al., also obtained floating gas-forming tablets using NaCMC in a mixture with HPMC K15M as matrix-forming substances. The obtained DF had an ascent delay time of 3.25 min, a flotation time of more than 12 h, and a sustained release of API, metronidazole [57].

### 5.4. HPC

HPC is a partially substituted poly hydroxypropyl cellulose ether, which is an excipient widely used in the technology of floating drug delivery systems, used as a matrix former, thickener in microencapsulation, and also as the main component of filaments in the production of DF using additive technologies [19,20]. Hyprolose is soluble to varying degrees in dichloromethane, ethanol, methanol, and propylene glycol, in water at a temperature below 38 °C (forms a colloidal solution), insoluble in hot water, the viscosity of the solutions depends on the grade, the bulk density of the polymer is approximately equal to 0.5 g/cm^3^ [19]. A feature of this polymer compared to other cellulose ethers, such as NaCMC and HPMC, is that it has the lowest true density and, accordingly, better buoyancy [20].

Giri et al., used HPC to prepare filaments together with stearic acid, from which then 3D-printed by fused deposition modeling (FDM) received hollow tablets of theophylline. The DF remained buoyant for 10 h, had no ascent delay, and possessed zero-order drug release kinetics [58]. Dumpa et al., obtained a floating DF based on the TiD (Tablet-in-Device) approach by making a core tablet containing theophylline, which was then placed in 3D-printed housing. FDM filaments are produced by hot extrusion from HPC and EC. The resulting tablets, due to the cavities in the framework, had a rise delay time equal to zero, a flotation time of up to 6 h, and a pulsatile release delay of 6 h [59]. Vo et al., obtained cinnarizine-loaded HPC-based filaments by hot extrusion, from which hollow tablets were subsequently made using FDM 3D printing. The resulting DF demonstrated instantaneous floating, a flotation time of more than 12 h, and zero-order release kinetics from 6 to 12 h, depending on the composition [60].

### 5.5. Other Semi-Synthetic Polymers

To a greater extent, the range of polymeric Es used in the field of obtaining floating dosage forms is limited to the examples described above. However, among the publications, there are examples of the use of other variants of semi-synthetic polymers.

Cellulose acetate (CA), a water-insoluble excipient used to prepare semi-permeable coatings or matrices for sustained-release tablets, was used by Bhardwaj et al., to obtain floating films loaded with 5-fluorouracil and a gas-forming agent. DF was placed in capsules, after the dissolution of which the films showed an ascent delay time of 85 to 142 s and a flotation time of 17 to 23 h, depending on the composition, providing a delayed release of API [19,61]. Ciprofloxacin microcapsules based on cellulose acetate were prepared by Asa and Mirzaeei by the solvent removal method. The resulting DF immediately surfaced, floated, and provided a sustained release of API for 24 h [62]. Hydroxyethyl cellulose is a non-ionic water-soluble polymer widely used as a coating material and a binder in the manufacture of tablets [19]. Kim et al., made an attempt to use this polymer as a matrix-forming agent together with HPMC, since compared to the latter, HEC shows faster and stronger swelling; however, in the manufacture of gas-forming floating tablets, the resulting gel could not withstand the stress of CO_2_ release [63].

## 6. Synthetic Polymers Used in the Technology of Floating Delivery Systems

Synthetic polymers in the technology of floating delivery systems are represented by aliphatic ethers, polymethacrylates, carbomers, and some other substances. In this area, Es of the described group are less common than semi-synthetic ones, however, as a rule, they perform the same functions and are often used in combination with them to improve buoyancy, swelling, and API release rates.

### 6.1. Aliphatic Polyesters

Aliphatic polyesters are a group of synthetic homopolymers or copolymers of lactic acid, glycolic acid, glycolide, and hydroxycaproic acid, including polylactic acid and polycaprolactone. Polylactic acid or polylactide (PLA) is a thermoplastic polyester of 2-hydroxypropanoic acid [19]. This polymer is insoluble in water, soluble in various organic solvents, and melts at a temperature of 165–180 °C. The tensile strength depends on the molecular weight and varies from 35–85 MPa, while this substance has a relatively high specific gravity of 1.21–1.28 g/cm^3^ [19]. PLA in the FDF technology is used to obtain fibers for 3D printing or insoluble coatings; in addition, it can act as a mucoadhesive [2].

This polymer was used by Malik et al., to obtain nanofibers loaded with diacerein by electrospinning. The resulting DF had a low density due to air cavities and mucoadhesion properties, improved the solubility of the API, had no rise delay time, floated for more than three days, and provided a controlled release of the drug for 30 h [64]. The same approach was taken by Fu et al., to create a floating DF of riboflavin. The case in the form of a cylinder with a cavity was printed on a 3D printer using polylactide-based filaments; tablet cores with a K15M HPMC matrix were obtained by direct compression. The collected DF was retained in the stomach of the test animals (rabbits) and provided a sustained release of API for more than three days [65].

Polycaprolactone (PCL) is a polyester synthesized from ε-caprolactone, with a molecular weight of 80,000 to 150,000, insoluble in water, soluble in some organic solvents, has a melting point of 58–63 °C, and a tensile strength of 20–35 MPa [19]. PCL can be used to obtain floating systems in the form of films, microcapsules, and microspheres [19].

For example, Lee et al., used PCL together with PLA to obtain hollow microspheres containing fenofibrate and metformin, coated with shells also based on PCL, with fenofibrate or piroxicam in the composition. DF demonstrated immediate flotation, due to the inclusion of olive oil as a density-reducing agent, the presence of a cavity, the retention of buoyancy for more than 24 h, and the fact that it provided delayed release of API [66].

### 6.2. Polyethylene Glycol (PEG)

PEG is a water-soluble polymer of ethylene glycol, which, depending on the molecular weight, usually varying from 200 to 35,000, is a liquid or solid, the density of the substance depends on the brand and varies from 1.11 to 1.21 g/cm^3^ [19]. In FDF technology, it is used as a matrix-forming agent and an agent that provides adhesion to mucous membranes [1,2,3,19].

Vasvari et al., developed the composition and technology for obtaining a floating metronidazole delivery system by melt foaming. PEG 4000, due to its polymeric nature, provided a high air capture during the mixing of the components, and the formation of a matrix with stearic acid, which provides a delayed release of the API. The resulting DF had no ascent delay, retained buoyancy for 10 h, and released metronidazole in accordance with the Korsmeier-Peppas kinetic model [67]. Haimhofer et al., developed a gastroretentive delivery system for acyclovir, which is a foam obtained by foaming with ultrasound (Ultrasonic Batch Technology). The matrix of this DF was formed by a porous mass of a solidified mixture of PEG 4000 and stearic acid. The foam obtained in this way had mucoadhesive properties, ensuring the absence of a rise delay, and flotation during the release in accordance with the zero-order kinetics model within 10 h [68].

### 6.3. Polyethylene Oxide (PEO)

PEO are non-ionic ethylene oxide homopolymers, soluble in water and some organic solvents, melting at a temperature of 65–70 °C [19]. The true density of these polymers is approximately 1.3 g/cm^3^ [19]. In the technology for obtaining floating delivery systems, PEO grades with high molecular weight are used as matrix formers; in addition, the described excipients have exceptional mucoadhesive properties and can also significantly increase in size upon swelling [1,2,3,19].

Jagdale et al., used PEO (Polyox WSR205 and Polyox WSR N12K) to create pulsating bisoprolol tablets. A mixture of these polymers with blowing agents was part of the molded coating deposited on the previously obtained core. The tablets floated to the surface of the dissolution medium within 3 min, remained buoyant for 9 h, and delayed the pulsatile release up to 4 h [69]. PEO samples with molecular weights ranging from 1,000,000 to 7,000,000 were used by Cvijic et al., to obtain floating gas-forming ranitidine tablets. Manufactured tablets of various compositions had an ascent delay time of 90 to 150 s, a flotation time of 90 to 640 min, and a delayed release of API [70].

### 6.4. Polymethacrylates

Polymethacrylates are synthetic cationic and anionic polymers of dimethylaminoethyl methacrylates, methacrylic acid, and methacrylic acid esters in various ratios [13,19]. Polymethacrylates are used to obtain film coatings and matrix components that provide modified release, as mucoadhesive agents, and are used to obtain microspheres by solvent removal [2,3,13,19].

Copolymer of methacrylic acid and methyl methacrylate (1:2) (Eudragit S-100) was used by Bansal et al., to obtain hollow microspheres of itopride hydrochloride by the method of solvent removal. DF had a zero ascent delay, a flotation time of 24 h, in vivo gastroretention of more than 8 h, and a delayed release of API [71]. The composition and technology for producing famotidine hollow microspheres based on Eudragit S-100 were developed by Gupta et al., DF prepared by the solvent removal method had an immediate rise to the surface, a flotation time, and a sustained release of the API up to 20 h [72]. The combination of a methacrylic acid-methyl methacrylate copolymer (1:1) (Eudragit L100) and a type B ammonio methacrylate copolymer (Eudragit RS100) was used to prepare pantoprazole hollow microspheres by Gupta et al., The pharmaceutical composition for reducing density also included magnesium stearate. The resulting DF had immediate flotation, retained buoyancy, and sustained release for 12 h [73]. Hollow microspheres of carvedilol were obtained by solvent removal by Ekta et al., using acrycoat S 100 as a polymer. DF did not have a rise delay, was retained on the surface of the dissolution medium for 12 h, and provided a delayed release of API [74].

### 6.5. Carbomers

Carbomers are synthetic high molecular weight acrylic acid polymers crosslinked with either allyl sucrose or allyl esters of pentaerythritol [19]. These polymers swell significantly in water and glycerin, but do not dissolve, since they are three-dimensionally cross-linked microgels, in addition, they have low bulk density and density after compaction: 0.2–0.4 g/cm^3^ and 0.3–0.4 g/cm^3^, respectively [19]. In the technology of obtaining floating delivery systems, they are used as matrix formers and low-density Es, and are also used to impart mucoadhesive properties to DF and a significant increase in size [1,2,3,21,75,76].

Ma et al., prepared gabapentin microspheres based on carbomer (Carbopol 934) by the solvent removal method. Due to the presence of pores in the structure and the low density of the polymer used, the resulting DF had immediate flotation, was retained on the surface of the dissolution medium for 9 h, and provided a delayed release of API for 12 h [77]. A combination of carbomer (Carbopol 971P) and HPMC K4M was used by Wani et al., to obtain a matrix-forming mixture with losartan for filling capsules. Due to the swelling ability of both polymers, their low bulk density, and the ability of carbopol to form an insoluble network structure, a stable buoyant gel was formed. DF remained buoyant in the dissolution medium for more than 12 h and provided a delayed release of API during this time; in vivo, the capsule and the resulting gel were retained in the stomach for more than 8 h [78]. Kumar et al., used a mixture of carbomer (Carbopol 934) and HPMC to obtain gas-forming floating tablets containing calcium disodium salt of ethylenediaminetetraacetic acid. The resulting DF was able to maintain flotation and controlled release of API for up to 24 h; in vivo, the tablets were retained in the stomach for 6 h [79]. Fernandes and Rathnanand developed carvedilol and nicotinamide cocrystal gassing tablets using carbomer (Carbopol 934P), HPMC E50, and K4M to form a CO_2_ retention matrix. The resulting DF had an ascent delay time of 11 s, a flotation time of 14 h, and provided a delayed release of API for 12 h [80].

### 6.6. Polyvinyl Alcohol (PVA)

PVA is a water-soluble thermoplastic synthetic polymer with a molecular weight of 20,000–200,000 [19]. PVA melts at a temperature of 228 °C (fully hydrolyzed grades) or 180–190 °C (partially hydrolyzed grades), and the polymer density ranges from 1.19 to 1.31 g/cm^3^ [19]. It is used in the technology of obtaining FDF as a material for the manufacture of filaments for 3D printing, as well as for the production of microspheres together with other Es [19].

Huanbutta and Sangnim have developed metronidazole floating tablets based on the TiD principle. The PVA case printed on a 3D printer was a cylinder with an air cavity and a cell for the core, at the bottom of which a special pore was made to slow down the transition of the API into the dissolution medium. The resulting dosage form did not have an ascent delay, floated for more than 4 h, and provided a controlled release of metronidazole for 8 h [81].

### 6.7. Other Synthetic Polymers

In addition to the Es described above, which are widely used to obtain various DFs, researchers often experiment with various materials that are in demand in technology in other areas of science. For example, Kulinowski et al., used fiber from a mixture of a thermoplastic polymer of nylon and carbon fiber (Nylon 12), which is widely used for 3D printing by selective laser sintering (SLS), to create a metronidazole delivery system. The porous insoluble matrix formed by nylon was a network structure with incorporated API. The resulting DF had a zero rise time delay, a flotation time of 30 to 600 min, and released metronidazole in a delayed manner over 0.5 to 5 h [82]. The removal method of Kumaraswamy et al., obtained porous floating microspheres, the matrix of which consisted of water-insoluble polycarbonates and polypropylene glycol, which provided an increase in the release of repaglinide into the aquatic environment. The resulting DF had no ascent delay, retained buoyancy for 12 h, and provided the release of API for 8 h in accordance with the Korsmeier-Peppas kinetics model [83]. A non-standard approach to selecting low-density Es to achieve flotation was used by Treesinchai et al., when creating floating tablets using expanded polypropylene powder (Accurel^®^ MP1004). The polymer retained by the HPMC matrix provided a zero rise delay time and a flotation time of more than 8 h [14]. Polyacrylamide together with HPMC E15 LV, PEG 6000, and Carbomer (Carbopol 934P) was used in the formulation developed by Rajput et al., floating and mucoadhesive DF of clarithromycin (“fanicular cylindrical system”). The resulting system provided delayed release of API for 8 h and floated due to swelling for 3 h [84].

All of the above polymers can be used individually or together in various combinations to achieve optimal DF performance (Table 1). Moreover, the properties of these Es allow them to be used to achieve flotation by simultaneously implementing several approaches to achieve flotation in one delivery system (Table 2).

## 7. Promising Directions in the Field of Obtaining Floating Dosage Forms and Polymers Used in These Approaches

The most effective approach to achieving buoyancy in terms of preventing evacuation to the underlying gastrointestinal tract is to obtain FDF with an initially low density. Immediate flotation, characteristic of such systems, prevents the removal of the drug from the stomach the first time after ingestion. Both approaches, the incorporation of air and low-density Es, provide equally effective targeted delivery of APIs, but the second option allows the use of classical technologies to obtain the most common dosage forms, such as tablets, capsules, and pellets, without including additional stages in the process, for example, sublimation, lyophilic drying or sonication. However, systems for which these steps are typical, in particular in microencapsulation, or DF obtained by methods that initially involve the formation of cavities or pores, for example, by hot extrusion or 3D printing, also have no obstacles to expanding the scope. On the other hand, the most common and market-successful approach to achieving buoyancy is the introduction of gas-forming substances or mixtures into the composition, as can be seen from the number of research papers and MP on the market [1,2,3]. This state of affairs is due to the ease of introducing gasifiers into tablets, one of the two most popular DFs used to create floating delivery systems, which is also explained with a large amount of practical data providing a basis for new work in this area. Other most widely used forms for the delivery of APIs to the stomach are microspheres and granules, due to the fact that they belong to multicomponent systems known for their ability to avoid the all-or-nothing effect characteristic of single-component dosage forms that occurs when they are destroyed or prematurely evacuated to the lower departments of the gastrointestinal tract [3]. In addition, the systems mentioned above have immediate flotation or a short ascent delay time, which is a significant advantage.

The frequency of application of certain Es in the development of floating delivery systems is directly related to the convenience of the production technology and the effectiveness of individual approaches to achieve flotation in the production of certain most popular dosage forms. Cellulose derivatives, especially HPMC, carbomers, polymethacrylates, and polyethylene oxides, are widely used in the creation of tablets or other compressed drugs. In the production of granules and microspheres by ionotropic gelation, polymers of natural origin are almost entirely used, capable of reacting with polyvalent metal ions, such as sodium alginate, petins, chitosan, in the case of solvent removal technology, semi-synthetic and synthetic Es are widely used, mainly ethyl cellulose and polymethacrylates.

The development of new production methods requires the use of specific materials. The current trend in the field of DF technology is the use of additive technologies (3D printing), which are associated with the possibility of obtaining personalized MP, achieving modified release, and other additional properties [151]. The main method for obtaining FDF is extrusion (FDM) printing, which is used to implement one of 2 concepts–manufacturing a tablet directly from API-loaded filaments or a TiD device, which is a printed case in which a tablet is placed. The most commonly used materials in these methods are PLA, HPC, HPMC, PVP, PVA, and PEG.

On the other hand, research on new materials in MP technology is moving towards obtaining Es with improved or additional properties, searching for more environmentally friendly excipients or using previously unknown sources of raw materials, as well as adapting substances used in other areas of science. In the field of gastroretentive systems, a combinatorial approach is important, in which the DF is given several properties that allow it to be retained in the stomach, most often, such as flotation, mucoadhesion, and a significant increase in size, which requires the use of several excipients or co-process polymers, or Es combine the ability to form a matrix, adhere to the surface of mucous tissues and swell. The ability to adhere and form a stable gel combines sodium alginate, chitosan, xanthan, gum, pectin, PEG, PLA, polymethacrylates. The property of significantly increasing in size is possessed by such a matrix former as NaCMC. All these characteristics are combined by such Es of a polymeric nature as: guar gum, HPMC, PEO, carbomers, and PLA. At present, the use of such natural excipients as the previously mentioned alginates, chitosan, and others is widespread, however, many works are devoted to the isolation and description of the technological properties of new plant polysaccharides, which are not inferior in their characteristics to already known materials, or the use of substances of animal origin, for example, hyaluronic acid, ovalbumin, etc. In addition, polymers that are not traditionally used in pharmacy, for example, nylon, polycarbonate, polystyrene, can be used as Es to obtain dosage forms, including floating ones.

## 8. Conclusions

Floating drug delivery systems make it possible to expand the possibilities of oral dosage forms and the range of APIs used in this way by providing targeted delivery to the stomach and their modified release, thereby improving bioavailability and reducing losses of the MP used, partially eliminating its side effects. The most important role in the technology of floating dosage forms is played by polymeric Es, traditionally subdivided on the basis of their origin, the functional properties of which ensure the achievement of the distinctive qualities of these systems: gastroretention, due to flotation, and modified release. Such characteristics are the ability to significant swelling, the formation of stable gels, low density, etc.

As a result of the analysis of the sources, the most promising and effective approaches to achieving buoyancy in the development of the FDF were identified. One-component gas-forming systems have acquired the greatest practical importance, the technology of which most often involves such semi-synthetic and synthetic polymers as HPMC, NaCMC, carbomers, and PEO, due to their ability to form matrices that can hold their shape for a long time, erode, maintain controlled release of API, and do not collapse under the influence of a rapidly ongoing reaction of carbon dioxide formation. Multicomponent DFs are also common, floating due to the presence of pores, cavities, or low-density Es in the matrix, widely used polymers in the preparation of which are NA, chitosan, and pectin, when using the ionotropic gelation method, and EC and polymethacrylates, in the case of using solvent removal, which is capable of forming strong, water-insoluble scaffolds.

The use of 3D printing methods has become an innovative direction of development in the field of technology for obtaining FDF. Based on the number of publications, the leading method is extrusion (FDM) printing, through which floating delivery systems based on API-containing filaments or TiD devices are obtained. Various thermoplastic polymers are widely used for this method, which is also acceptable for use in pharmacy, such as HPC, PLA, PEG, PVA, and PVP. In addition, there is a tendency to adapt other additive technologies and materials used in them for the described purpose, for example, there is work on the use of SLS printing to obtain nylon-based floating tablets.

A promising area is the introduction of materials not previously widely used in pharmacy as excipients, as a rule, substances that are hydrophilic polymers, such as hyaluronic acid, egg albumin, bovine albumin, or, conversely, insoluble, for example, polycarbonates, polypropylene, etc. At the same time, a large number of works are devoted to obtaining fundamentally new Es, for example, polysaccharides isolated from plant materials or polymers, by synthesizing graft copolymers based on natural and synthetic components.

## Figures and Tables

**Figure 1 pharmaceutics-14-02779-f001:**
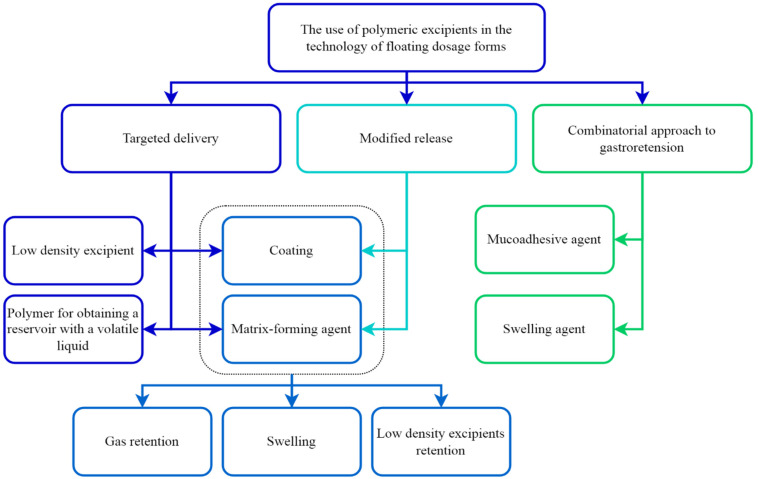
Field of application of polymer excipients in the technology of floating dosage forms.

**Figure 2 pharmaceutics-14-02779-f002:**
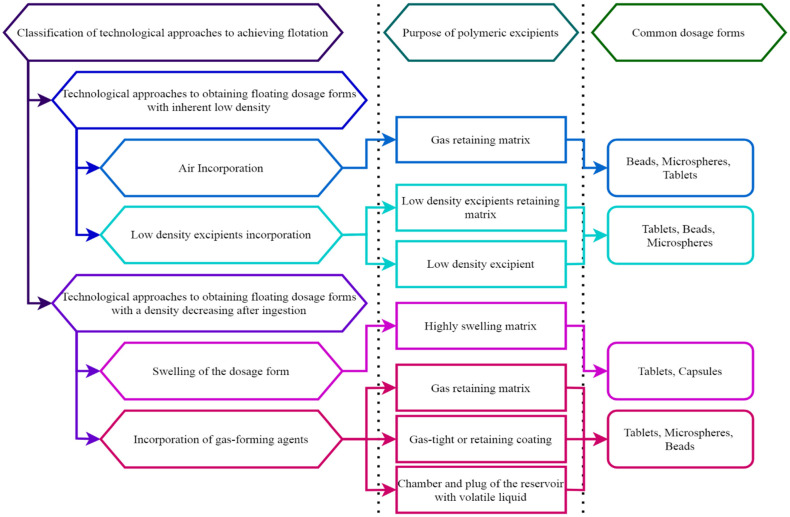
Classification of technological approaches to achieve flotation, the purpose of polymers in these approaches, and the most common dosage forms.

**Table 1 pharmaceutics-14-02779-t001:** Examples of the use of various polymers in the field of obtaining floating drug delivery systems.

Dosage Form and Production Technology	Polymers and Other Es and Their Purpose	Ref.
**Air Incorporation**		
Hollow granules (ionotropic gelation)	Chitosan	[25]
Granules (ionotropic gelation)	NA, Carrageenan	[28]
	NA, *Eudragit RL100*	[85]
	Chitosan	[26]
	Pectin	[30]
	NA conjugate with bovine serum albumin	[37]
Hollow microspheres (solvent removal)	EC	[47]
	EC, HPMC E5	[50]
	EC, HPMC E50	[86]
	*Eudragit S-100*	[71]
	*Eudragit S-100*	[72]
	*Acrycoat S 100*	[74]
	Combination of *Eudragit L100* and *Eudragit RS100*	[73]
	Polycarbonate, Polypropylene Glycol ^PFA^	[83]
	HPMC 3K, *Eudragit RS PO*	[87]
	EC (10 cP), PVP	[88]
Microspheres (solvent removal)	EC	[89]
	EC, HPMC K4M	[90]
	CA	[62]
Microspheres (“multiple emulsion method”, solvent removal)	EC, PVP	[91]
Tablets (direct compression)	HPMC K4M, Camphor ^SA^	[55]
Tablets (wet granulation)	Locust bean gum, Camphor ^SA^	[41]
	HPC (*Klucel LF*), PEO (*Polyox WSR N 60 K*), Menthol ^SA^	[92]
Tablets (shaping and freeze drying)	EC, HPMC (100 cP)	[49]
Tablets (FDM 3D printing)	HPC	[60]
	HPMC acetate succinate, PEG 400	[93]
	HPC, PVP	[94]
Tablets (Tablet-core–direct compression; body–FDM 3D printing)	HPC, EC	[59]
	PLA	[81]
	EVOH, HPMC K15M	[65]
Tablets (SLS 3D printing)	Nylon	[82]
Cylindrical extrudate (3D printing with simultaneous ionotropic gelation)	NA	[23]
Hydrogel (ionotropic gelation)	Graft copolymer of chitosan and PVP	[27]
Nanofiber (electrospinning)	PLA	[64]
Pellets or cylindrical extrudates (extrusion)	*Eudragit RSPO*, HPMC K15 M, Ethanol ^PFA^	[95]
Modular gastro-renal delivery system (wet granulation and compression)	HPMC K15 M, PEG 6000, PVP K30	[96]
**Incorporation of excipients with low density**		
Tablets (direct compression)	HPMC K4M, Glyceryl behenate ^LDE^	[54]
	HPMC K4M, HPMC K15 M, HPMC K100LV, Polypropylene ^LDE^	[14]
	HPMC K100 M, Glyceryl behenate ^LDE^	[97]
	HPMC K100 LV, PEO (*Polyox 301WSR*), Functionalized calcium carbonate ^LDE^	[98]
Tablets (core–melt spinning, extruded coating)	PEO 5,000,000, PEG 6000, Stearic acid ^LDE^	[99]
**Incorporation of blowing agents**		
Tablets (direct compression)	Xanthan gum	[32]
	HPMC K 15M, NA	[100]
	Egg albumin	[46]
	EC, HPMC K4M	[48]
	HPMC K4M, HPMC K100M, HPMC K100LV	[52]
	HPMC K4M	[53]
	NaCMC (2000 cP), HPMC K4M	[56]
	HPMC K4M	[101]
	HPMC E 15LV	[102]
	HPMC K4 M, HPMC E5LV	[103]
	HPMC K100 M CR	[104]
	PEO (molecular mass: 1,000,000–7,000,000)	[70]
	Carbomer (Carbopol 934), HPMC K4M	[79]
	Carbomer (Carbopol 934P), HPMC K4M, HPMC E50	[80]
	HPMC K15 M, Carbomer (*Carbopol 971 P NF*), NaCMC	[105]
	HPMC K15 M, EC, Gum	[106]
	HPMC K4 M, Carbomer (*Carbopol 974P*) ^MA^	[107]
	HPMC K100 M, Mucus secreted from *Hibscus rosa-sinensis*, Carbomer (*Carbopol 934P*) ^MA^	[108]
	HPMC K4 M, HPMC K15 M, PEO (*Polyox 303WSR*)	[109]
	NaCMC, PEO (*Polyox 303WSR*), Carbomer (*Carbopol 934P*)	[110]
	HPMC K4 M, Locust bean gum	[111]
	EC, HPMC K100 M	[112]
	HPMC K15 M, Carbomer (*Carbopol 971P*)	[75]
	NA, HPC	[113]
	Carbomer (*Carbopol 934P*), NA, HPMC K4 M, PVP	[114]
	HPMC K4 M, PVP K30	[115]
	Graft copolymer of polyacrylamide and corn fiber gum ^MA^	[116]
	Carbomer (*Carbopol 934P*), PEO (*Polyox N-60K*)	[117]
Tablets (Direct Compression, Compressed Coatings)	HPMC K4M, HPMC E50	[51]
	PEO (*Polyox WSR205; Polyox WSR N12K*)	[69]
	PEO (*Polyox WSR205*), Xanthan gum	[118]
	HPMC K100 M, Carbomer (*Carbopol 971P*)	[119]
Tablets (wet granulation)	Co-processing aid based on xanthan gum and acacia gum	[33]
	Xanthan gum, a polysaccharide isolated from *Orchis morio*	[43]
	Xanthan Gum, Polysaccharide from *Tamarindus indica* seeds	[43]
	NaCMC, HPMC K15 M	[57]
	Carbomer (Carbopol 940), HPMC K4M	[120]
	HPMC K15 M, HPMC K100 M	[121]
	Carbomers (Carbopol 934P; 971 P) HPMC K4M	[122]
	HPMC K4 M, Carbomer, Chitosan	[123]
	EC, Eudragit RS PM	[124]
	NA, HPMC K4 M, Carbomer (*Carbopol 974P*)	[125]
	Polysaccharide from *Tamarindus indica* seeds, HPMC K4 M, PVP	[126]
	HPMC K4 M, NA, Carbomer (*Carbopol 934P*), PEG 3500	[127]
Tablets (wet granulation, film coating)	HPMC K4 M	[128]
Tablets (wet granulation, pressed coatings)	HPC, NA	[119]
	HPMC K15 M, PEO (*Polyox 303 SWR*)	[129]
Bilayer tablets (wet granulation)	Eudragit RL 100, CMC	[130]
Tablets (dry granulation)	HPMC K4	[131]
Minitablets (wet granulation) encapsulated	HPMC K100 M	[132]
Granules or “spheroids” (extrusion-pelletization)	HPMC K4 M, Carbomer (*Carbopol 934P*)	[133]
Film (solvent removal) encapsulated	CA	[61]
**Incorporation of swelling substances**		
Tablets (direct pressing)	Chitosan salts ^SE^, HEC	[134]

**Note:** PFA—pore-forming agent; SA—sublimating agent; LDE—low density excipient; SE—swelling excipient; MA—mucoadhesive agent.

**Table 2 pharmaceutics-14-02779-t002:** Examples of the use of various polymers in the field of obtaining floating drug delivery systems by combining several approaches to achieve flotation.

Dosage Form and Production Technology	Polymers and Other Es and Their Purpose	Ref.
**Incorporation of air and excipients with low density**		
Granules (ionotropic gelation)	NA, Myristyl alcohol ^LDE^	[22]
	NA, Olive oil ^LDE^	[135]
	NA, Olive oil ^LDE^	[24]
	NA, Chitosan, Olive oil ^LDE^	[136]
	NA, *Cremophor EL* ^LDE^	[137]
	Pectin, Glyceryl monooleate ^LDE^	[31]
	NA, Chitosan ^MA^, Magnesium stearate ^LDE^	[138]
Granules (gelatinization of emulsion)	NA, EC, Isopropyl myristate ^LDE^	[139]
	NA, Light mineral oil ^LDE^	[140]
Hollow microspheres (solvent removal, spray drying coating)	PCL, PLA, Olive oil ^LDE^	[66]
Microspheres (solvent removal)	Carbomer (Carbopol 934) ^LDE^	[77]
Microspheres (granulation, solvent removal)	EC, *Eudragit RS100*, HPMC E5 ^LDE^	[141]
Tablets (FDM 3D printing)	HPC, Stearic acid ^LDE^	[58]
Mini-tablets with different structures (FDM 3D printing)	Ethylene vinyl acetate copolymer ^LDE^, Copolymer of vinylpyrrolidone, and vinyl acetate, EVOH	[142]
Foam (foaming melt)	PEG 4000, Stearic acid ^LDE^	[67]
	PEG 4000, Stearic acid ^LDE^	[143]
Foam (foaming by Ultrasonic Batch Technology)	PEG 4000, Stearic acid ^LDE^	[68]
Pills (extrusion)	Hyaluronic acid, Cetyl alcohol ^LDE^	[36]
Pellets (extrusion-pelletizing)	Hydroxyethylcellulose, HPMC (6 cP), Glyceryl behenate ^LDE^	[144]
**Incorporation of air and blowing agents**		
Hollow granules (ionotropic gelling)	NA, *Eudragit RSPO*, Eudragit L100	[145]
Granules (ionotropic gelation)	Pectin isolated from banana peel (*Musa balbisiana ABB*)	[35]
	Pectin, Xanthan gum, Locust bean gum	[39]
Granules (“extrusion curing method”)	NA, HPMC	[146]
Microspheres (ionotropic gelation)	NA, Chitosan ^MA^, Mucus isolated from *Mimosa pudica* seeds ^MA^	[38]
	Iota Carrageenan, NA	[29]
Extrudates (extrusion)	Carbomer (*Carbopol 934*), HPMC, *Eudragit RS100*	[147]
**Incorporation of air and swelling substances**		
Bilayer tablets (direct compression) The floating layer of tablets is based on sodium alginate granules	NA, PEG 6000, HPMC K100 M, PEO (*Polyox 303WSR*) ^SE^, Carbomer (*Carbopol 971P*) ^SE^	[148]
**Incorporation of swelling agents and excipients with low density**		
“Fanicular cylindrical system” (molding)	Polyacrylamide, HPMC E15 LV, PEG 6000, Carbomer (*Carbopol 934P*) ^LDE, SE^	[84]
**Incorporation of blowing agents and swelling agents**		
Tablets (direct pressing)	Xanthan gum ^SE^, Glucomannan ^SE^, HPMC K15 M ^SE^, Polysaccharide from *Tamarindus indica* seeds ^SE^	[149]
**Incorporation of low-density blowing agents and excipients**		
Tablets (melt granulation)	Graft copolymer of polyacrylamide and *Azadirachta indica* gum*,* Carnauba wax ^LDE^	[150]
**Incorporation of air, swelling agents, and low-density auxiliary agents**		
Capsules (capsule filling)	Carbomer (Carbopol 971P) ^LDE, SE^, HPMC K4M ^SE^	[78]

**Note:** LDE—low-density excipient; SE—swelling excipient; MA—mucoadhesive agent.

## Data Availability

Not applicable.

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
