# Peer review of "Polymeric Excipients in the Technology of Floating Drug Delivery Systems"

_pharmaceutics, 2022, doi:10.3390/pharmaceutics14122779_

Round 1

Reviewer 1 Report

The manuscript, "Polymeric Excipients In The Technology Of Floating Drug Delivery Systems" reviews about the use of excipients and polymers, natural and synthetic, and targeted approach in floating drug delivery systems. Moreover, the figures adequately explain the application of polymers in this aspect along with the classifications. The tables adequately explains the floating systems, different excipients and polymers used in the development of floating oral systems. 

The manuscript is easy to understand and can be of interest to wide audience. 

Author Response

Thank you so much for your positive opinion.

Reviewer 2 Report

The manuscript entitled "Polymeric Excipients In The Technology Of Floating Drug Delivery Systems" is a review article regarding the role of various polymeric compounds in the preparation of orally administrated drug formulations that have the ability to float into the stomach. In general, the review is interesting, it is written well, easy to follow and the references are updated. My only minor comment to the authors is to include a section describing the methodology used for the preparation of the review (i.e., time period for papers included in the review, keywords used, no of articles found and the  inclusion/exclusion criteria employed etc.).

Author Response

To Reviewer 2

Dear Reviewer,

We seriously considered your comment (below):

  • “My only minor comment to the authors is to include a section describing the methodology used for the preparation of the review (i.e., time period for papers included in the review, keywords used, no of articles found and the  inclusion/exclusion criteria employed etc.).”

and included the methodology of articles of interest selection in the submission.

Thank you so much for your valuable comments

Regards

Alex

Reviewer 3 Report

The manuscript is well structured. In my opinion, it meets the expectations of the newspaper. The summary provides great help for the development of floating capsules later.

Author Response

(The authors gave the same response as above.)
